

# Uric acid to HDL cholesterol ratio as a novel predictor of carotid intima-media thickness: a cross-sectional study in rural China

Guibao Luo[1,*], Haiying Wang[2,*], Yajun Cao[3], Juan Hao[1], Taofeng Tan[1], Yingzhe Shao[1], Xianjia Ning[1,4,5,6], Chunsheng Yang[1], Jinghua Wang[1,4,5,6] and Yan Li[5]

[1] Department of Neurology, Tianjin Medical University General Hospital, Tianjin, China
[2] Department of Cardiology, Tianjin Jizhou People's Hospital, Tianjin, China
[3] Tianjin Jizhou District Chuanfangyu Township Hospital, Tianjin, China
[4] Tianjin Neurological Institute, Key Laboratory of Post-Neuroinjury Neuro-repair and Regeneration in Central Nervous System, Ministry of Education and Tianjin City, Tianjin, China
[5] Institute of Clinical Epidemiology & Evidence-Based Medicine, Tianjin Jizhou People's Hospital, Tianjin, China
[6] Laboratory of Epidemiology, Tianjin Neurological Institute, Tianjin, China
* These authors contributed equally to this work.

Corresponding authors
Jinghua Wang, jwang3@tmu.edu.cn
Yan Li, liyanmanu@163.com

## ABSTRACT

**Purpose:** Carotid intima-media thickness (cIMT) is a widely recognized marker for assessing carotid atherosclerosis, which is a significant predictor of cardiovascular diseases such as ischemic stroke. The serum uric acid to high-density lipoprotein cholesterol ratio (UHR) has recently emerged as a potential combined marker for metabolic and inflammatory processes related to cardiovascular risk. This study aims to investigate the association between UHR and cIMT in a rural Chinese population, with a particular focus on differences across age and sex groups.

**Patients and methods:** A cross-sectional study was conducted using data from the 2019 general physical examination of residents in Jizhou District, Tianjin, China. A total of 3,280 participants aged 45 years and older were included after excluding those with incomplete data or specific health conditions. Demographic and clinical data were collected through face-to-face interviews and physical examinations. Serum levels of uric acid (SUA) and high-density lipoprotein cholesterol (HDL-C) were measured, and UHR was calculated. Carotid intima-media thickness was assessed using ultrasound. Multivariate linear and logistic regression analyses were performed to evaluate the association between UHR and cIMT, adjusting for potential confounders. Subgroup analyses were conducted to explore variations in this association by age and sex. Receiver operating characteristic (ROC) analysis was used to compare the predictive value of UHR, SUA, and HDL-C for cIMT thickening.

**Results:** The study population had a mean age of 64.10 ± 8.02 years, with 45.2% males and 54.8% females. The prevalence rates of hypertension, diabetes, smoking, and alcohol consumption were 77.2%, 19.7%, 41.4%, and 36.3%, respectively. Univariate analysis showed significant associations between UHR, age, hypertension, diabetes, smoking, alcohol consumption, systolic blood pressure (SBP), pulse pressure (PP) difference, glucose (GLU), total cholesterol (TC), low-density lipoprotein cholesterol (LDL-C), and cIMT. Multivariate analysis revealed that UHR

was an independent risk factor for increased cIMT ($\beta$ = 0.06, 95% CI [0.01–0.10], $P$ = 0.017) and carotid intima-media thickening, particularly in older men. Subgroup analysis indicated that the association between UHR and cIMT was more pronounced in participants aged 60 years or older and in males. ROC analysis demonstrated that UHR had a higher predictive value for cIMT thickening in older men (AUC = 0.577, 95% CI [0.510–0.644], $P$ < 0.05) compared to SUA or HDL-C alone.

**Conclusion:** This study identifies UHR as a significant predictor of cIMT and carotid intima-media thickening, with a particularly strong association observed in older men. These findings suggest that UHR could serve as a valuable marker for early detection and intervention of carotid atherosclerosis.

## INTRODUCTION

Globally, stroke is the second leading cause of death, accounting for 11.6% of all fatalities, and the third leading cause of death and disability, contributing to 5.7% of total disability-adjusted life years (DALYs). The age-standardized stroke-related mortality rate in low-income countries is 3.6 times higher than in high-income countries, and the age-standardized stroke-related DALY rate is 3.7 times higher (*GBD 2019 Stroke Collaborators, 2019*). As the largest developing country, China has the highest number of stroke patients worldwide (*Tu, Wang & The Special Writing Group of China Stroke Surveillance Report, 2023*). In China, stroke is the leading cause of death and disability, with the burden of the disease rapidly increasing. In 2020, the overall prevalence, incidence, and mortality rates of stroke among adults aged 40 years and older were estimated at 2.6%, 505.2 per 100,000 person-years, and 343.4 per 100,000 person-years, respectively (*Tu et al., 2023*). Stroke-related DALYs are higher than those for most other diseases, including heart disease, respiratory disease, and digestive disease (*Wu et al., 2019*). Ischemic stroke, which accounts for 86.8% of all stroke events, has three primary causes: 50% are due to cerebrovascular atherosclerotic plaques and plaque rupture, 20% are caused by cardiac cerebral infarction, and 25% result from lacunar cerebral infarction due to small-vessel lesions (*Warlow et al., 2003*). Carotid intima-media thickness (cIMT) is a non-invasive imaging marker of carotid atherosclerosis that enables the early identification of atherosclerotic vascular changes, which can be used to assess the risk of cardiovascular and cerebrovascular diseases and serves as a strong predictor for their occurrence (*Lorenz et al., 2007*). It has emerged as a key tool for assessing and monitoring the initiation and progression of cardiovascular and cerebrovascular diseases (*Nezu et al., 2016*).

Recent studies have identified serum uric acid (SUA) as a risk factor for atherosclerosis (*Feig, Kang & Johnson, 2008*). High-density lipoprotein cholesterol (HDL-C) is known for its protective effects against atherosclerosis through anti-inflammatory and antioxidant mechanisms (*Linton et al., 2023*). The serum uric acid to high-density lipoprotein

cholesterol ratio (UHR) has emerged as a novel inflammatory and metabolic marker with potential predictive value for ischemic heart disease, metabolic syndrome, and non-alcoholic fatty liver disease (*Park, Jung & Lee, 2022*; *Yazdi et al., 2022*; *Xie et al., 2023*). Despite extensive research on SUA and HDL-C, the association between UHR and carotid atherosclerosis, specifically cIMT, remains underexplored.

Several controversies and gaps persist in this research area. While some studies suggest a positive correlation between elevated SUA and increased cIMT, others do not find significant associations (*Strasak et al., 2008*; *Jiménez et al., 2016*). The role of HDL-C in preventing atherosclerosis is well-documented, yet the combined effect of SUA and HDL-C, represented by UHR, on carotid atherosclerosis is not fully understood. Furthermore, the impact of UHR on carotid intima-media thickening, particularly in different demographic groups, has not been thoroughly investigated.

This study aims to elucidate the relationship between UHR and cIMT in a rural Chinese population. By conducting a retrospectively cross-sectional analysis, we seek to determine whether UHR can serve as a reliable marker for early detection and intervention of carotid atherosclerosis. We hypothesize that a higher UHR is associated with increased cIMT and that this association may vary across age and sex.

## METHODS

### Study population

The data for this retrospectively cross-sectional study were collected from the general physical examination records of residents in rural Jizhou District, Tianjin, China, in 2019. The inclusion criteria were: (1) age 45 years or older; (2) ability to cooperate with questionnaires and physical examinations, excluding those with major diseases, traumatic injuries, congenital diseases, or psychiatric disorders; (3) no history of cardiovascular and cerebrovascular diseases, including acute myocardial infarction, ischemic stroke, or hemorrhagic stroke. Exclusion criteria included individuals with viral hepatitis, cirrhosis, chronic kidney disease (CKD ≥ stage 4), pregnant women, and those in a paralyzed state. The study was conducted in accordance with the Declaration of Helsinki and was approved by the Ethics Committee of the General Hospital of Tianjin Medical University (approval number: IRB2018-100-01). All participants provided written informed consent.

### Data collection

Demographic and clinical data were collected through face-to-face interviews conducted by trained epidemiological researchers using pre-designed standardized questionnaires. Collected data included name, sex, age, lifestyle factors (smoking status and alcohol consumption), presence of diabetes and hypertension, and current use of antihypertensive and hypoglycemic medications. Participants were categorized into three age groups: 45–59 years, 60–69 years, and 70 years or older. Lifestyle information included smoking status (never smoked or smoked) and alcohol consumption status (never drank or drank).

## Physical and biochemical examination

Physical measurements included height (cm), weight (kg), and blood pressure (mmHg), all conducted in a fasting state. Blood samples were collected to measure fasting blood glucose (GLU) and serum concentrations of total cholesterol (TC), triglycerides (TG), high-density lipoprotein cholesterol (HDL-C), and low-density lipoprotein cholesterol (LDL-C). Carotid ultrasound was performed by trained physicians to measure carotid intima-media thickness (IMT). Body mass index (BMI) was calculated as weight in kilograms divided by the square of height in meters.

## Carotid ultrasound measurement

An ultrasound instrument equipped with electronic calipers was used to measure carotid IMT in real time. Trained sonographers performed the ultrasound without knowledge of participants' baseline data. The proximal and distal walls and bifurcations of the common carotid, internal carotid, and external carotid arteries were scanned in anterior, lateral, and posterior projections using a standardized protocol. Carotid artery intima-media thickening was defined as an intima-media thickness greater than 1.0 mm (*Lin et al., 2022*).

## Covariate Definitions

Hypertension was defined as a systolic blood pressure (SBP) value of $\geq$ 140 mmHg and/or a diastolic BP (DBP) value of $\geq$ 90 mmHg, self-reported hypertension, and/or the use of antihypertensive medications (*Xie et al., 2023*). Diabetes (particularly type 2 diabetes mellitus (T2DM)) mellitus was defined by one of the following criteria: self-reported diabetes, use of glucose-lowering medication or insulin, fasting glucose (GLU) value of 7 mmol/L or more, or hemoglobin A1c (HbA1c) value of 6.5% or higher (*Xie et al., 2023*). Smoking was defined as smoking at least one cigarette per day for more than 1 year. Alcohol consumption was defined as drinking alcohol at least once a week and having a total alcohol intake of more than 50 ml per week for more than 6 months. Participants were categorized into four BMI groups based on China-specific criteria: underweight (BMI < 18.50 kg/m$^2$), normal weight (18.50 kg/m$^2$ $\leq$ BMI < 24.00 kg/m$^2$), overweight (24.00 kg/m$^2$ $\leq$ BMI < 28.00 kg/m$^2$), and obese (BMI $\geq$ 28.00 kg/m$^2$) (*Jiang et al., 2016*).

## Statistical analysis

Normally distributed variables were expressed as mean $\pm$ standard deviation ($\bar{x} \pm s$). Differences between groups were compared using the t-test. Categorical variables were expressed as numbers (frequency), and differences between groups were compared using the chi-square test. Logistic regression analysis was employed for multivariate analysis of carotid intima-media thickness. Carotid intima-media thickness and carotid intima-media thickening were used as dependent variables, while UHR and other variables that showed statistical significance in univariate analyses were used as independent variables. A general linear model with robust standard errors was employed to assess the relationships between independent and dependent variables. The relationship between UHR and carotid intima-media thickness was expressed as an adjusted odds ratio (OR) and 95% confidence

interval (95% CI). A *P*-value of <0.05 was considered statistically significant. All statistical analyses were performed using SPSS software (version 27.0; IBM, Armonk, NY, USA).

## RESULTS

Initially, 3,641 participants were enrolled for the general physical examination in 2019. After excluding 64 participants under 45 years of age, 88 participants without complete uric acid and HDL-C cholesterol data, and 106 participants without complete carotid ultrasound data, the final study population comprised 3,280 participants (Fig. 1).

### Characteristics of the study participants

A total of 3,280 participants were included in the study, comprising 45.2% males and 54.8% females. The mean age was 64.10 ± 8.02 years. The prevalence rates of hypertension, diabetes, smoking, and alcohol consumption were 77.2%, 19.7%, 41.4%, and 36.3%, respectively. The mean BMI was 25.93 ± 3.71 kg/m$^2$, the mean IMT was 715.03 ± 142.06 μm, and the mean UHR was 239.73 ± 108.81 (Table 1).

### Univariate analysis of UHR and IMT

Univariate general linear model analysis showed that UHR, gender, age, hypertension, diabetes, smoking, alcohol consumption, systolic blood pressure, pulse pressure (PP) difference, GLU, TC, LDL-C, and SUA were significantly correlated with IMT ($P < 0.05$). Gender was negatively correlated with IMT, while all other factors were positively correlated with IMT (Table S1).

### Multivariate Analysis of UHR and IMT

In the multivariate general linear model analysis, UHR was found to be a significant risk factor for IMT. The analysis was conducted in three models. Model I adjusted for age and sex (β = 0.06, 95% CI [0.01–0.10], $P = 0.017$); Model II adjusted for age, sex, smoking history, and alcohol consumption (β = 0.05, 95% CI [0.01–0.10], $P = 0.022$); Model III adjusted for age, sex, smoking history, alcohol consumption, hypertension, diabetes, differential pulse pressure, GLU, TC, and LDL-C (β = 0.06, 95% CI [0.01–0.10], $P = 0.017$) (Table 2).

### Univariate analysis of UHR and carotid intima-media thickening

Univariate analysis revealed that UHR, age, hypertension, diabetes, smoking, alcohol consumption, systolic blood pressure, pulse pressure difference, GLU, and SUA were risk factors for carotid intima-media thickening ($P < 0.05$). Gender was the only protective factor for carotid intima-media thickening ($P < 0.05$) (Table S2).

### Multivariate analysis of UHR and the presence of carotid intima-media thickening

Multivariate logistic regression analysis showed that UHR was significantly associated with carotid intima-media thickening in Model I (β = 1.002, 95% CI [1.000–1.003], $P = 0.033$); Model II (β = 1.002, 95% CI [1.000–1.003], $P = 0.034$); however, in Model III, UHR was not correlated with carotid intima-media thickening (Table 2).
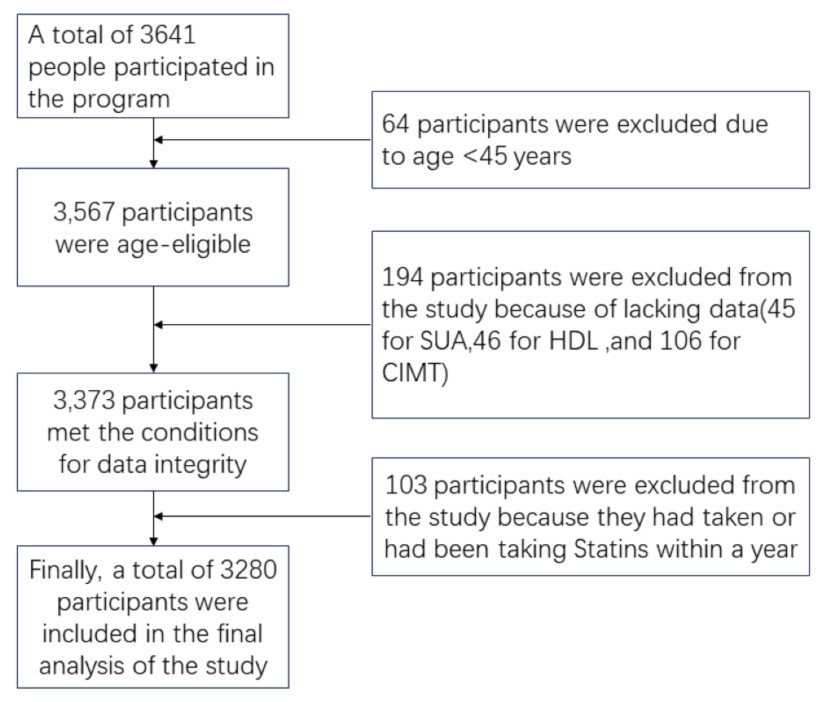

**Figure 1 Flow chart of participants' selection.**

**Table 1 Characteristics of the study participants.**

| Item | Male | Female | Total | P |
|---|---|---|---|---|
| N (%) | 1,484 (45.2) | 1,796 (54.8) | 3,280 | |
| Age, year, means ± SD | 65.06 ± 8.05 | 63.31 ± 7.91 | 64.10 ± 8.02 | <0.001 |
| Age groups, year, n (%) | | | | <0.001 |
| <60 | 320 (21.6) | 563 (31.3) | 883 (26.9) | |
| 60–69 | 762 (51.3) | 848 (47.2) | 1,610 (49.1) | |
| ≥70 | 402 (27.1) | 385 (21.4) | 787 (24.0) | |
| Hypertension, n (%) | 1,156 (77.9) | 1,376 (76.6) | 2,532 (77.2) | <0.001 |
| Diabetes, n (%) | 243 (16.4) | 404 (22.5) | 647 (19.7) | <0.001 |
| Smoking, n (%) | 1,242 (83.1) | 117 (6.4) | 1,359 (41.4) | <0.001 |
| Alcohol consumption, n (%) | 1,106 (74.5) | 83 (4.6) | 1,189 (36.3) | <0.001 |
| BMI[1], kg/m², means ± SD | 25.49 ± 3.49 | 26.29 ± 3.86 | 25.93 ± 3.71 | <0.001 |
| BMI groups, n (%) | | | | <0.001 |
| Normal weight | 505 (34.5) | 486 (27.5) | 991 (30.2) | |
| Overweight | 641 (43.7) | 771 (43.5) | 1,412 (44.4) | |
| Obese | 319 (21.8) | 514 (29.0) | 833 (25.4) | |
| Systolic blood pressure[2], mmHg, means ± SD | 151.55 ± 20.09 | 149.76 ± 19.90 | 150.57 ± 20.00 | 0.012 |
| Diastolic blood pressure[2], mmHg, means ± SD | 87.79 ± 10.94 | 83.13 ± 10.58 | 85.23 ± 10.99 | <0.001 |
| Differential pulse pressure[2], mmHg, means ± SD | 63.77 ± 15.90 | 66.63 ± 16.19 | 65.34 ± 16.12 | <0.001 |
| IMT, μm, means ± SD | 737.77 ± 148.90 | 696.24 ± 133.29 | 715.03 ± 142.06 | <0.001 |
| IMT group, n (%) | | | | <0.001 |
| <1,000 μm | 1,396 (94.1) | 1,760 (98.0) | 3,156 (96.2) | |

| Item | Male | Female | Total | P |
|---|---|---|---|---|
| ≥1,000 µm | 88 (5.9) | 36 (2.0) | 124 (3.8) | |
| GLU, mmol/L, means ± SD | 5.92 ± 1.66 | 6.02 ± 1.68 | 5.98 ± 1.67 | 0.066 |
| TC, mmol/L, means ± SD | 4.89 ± 0.90 | 5.28 ± 0.92 | 5.10 ± 0.93 | <0.001 |
| TG, mmol/L, means ± SD | 1.51 ± 1.51 | 1.73 ± 1.20 | 1.63 ± 1.35 | <0.001 |
| HDL-C, mmol/L, means ± SD | 1.34 ± 0.38 | 1.37 ± 0.34 | 1.35 ± 0.36 | 0.015 |
| LDL-C, mmol/L, means ± SD | 2.98 ± 0.82 | 3.25 ± 0.99 | 3.13 ± 0.93 | <0.001 |
| SUA, µmol/L, means ± SD | 325.94 ± 85.93 | 273.88 ± 77.45 | 297.44 ± 85.41 | <0.001 |
| UHR, means ± SD | 266.94 ± 114.61 | 217.25 ± 98.27 | 239.73 ± 108.81 | <0.001 |

Note:
Percentages are given in parentheses. UHR is the serum uric acid to high-density lipoprotein cholesterol ratio. GLU, TC, TG, LDL-C, HDL-C are in mmol/L, and uric acid is in µmol/L. A total of 19 males and 25 females were missing in ①; 51 males and 54 females were missing in ②.

**Table 2 Association of UHR with IMT and intima-media thickening in the multifactor analysis.**

| Item | β (95% CI) | P |
|---|---|---|
| IMT[a] | | |
| Model I | 0.06 [0.01–0.10] | 0.017 |
| Model II | 0.05 [0.01–0.10] | 0.022 |
| Model III | 0.06 [0.01–0.10] | 0.017 |
| Intima-Media Thickening[b] | | |
| Model I | 1.002 [1.000–1.003] | 0.033 |
| Model II | 1.002 [1.000–1.003] | 0.034 |
| Model III | 1.001 [1.000–1.003] | 0.061 |

Notes:
[a] Model I: adjusted for age, sex; Model II: adjusted for age, sex, smoking history, alcohol consumption; Model III: adjusted for age, sex, smoking history, alcohol consumption, hypertension, diabetes, differential pulse pressure, GLU, TC, LDL-C.
[b] Model I: adjusted for age, sex; Model II: adjusted for age, sex, smoking history, alcohol consumption; Model III: adjusted for age, sex, smoking history, alcohol consumption, hypertension, diabetes.

## Subgroup analysis of the association between UHR and carotid intima-media thickening

Subgroup analyses stratified by age and sex showed that UHR was significantly associated with carotid intima-media thickening in participants aged 60 years or older and in the male population (Table 3). Further analysis combining age and sex indicated that UHR was significantly associated with carotid intima-media thickening in older men (Table S3). Specifically, in elderly men, each unit increase in UHR increased the likelihood of carotid intima-media thickening by 0.2% (OR = 1.002, 95% CI [1.000–1.004], $P = 0.028$) (Table 4).

## ROC analysis

Receiver operating characteristic (ROC) analysis demonstrated that in older men, UHR had a higher predictive value for carotid intima-media thickening than SUA or HDL-C alone. The area under the curve (AUC) for UHR was 0.577 (95% CI [0.510–0.644], $P < 0.05$) (Fig. 2). After calculation, the optimal truncation value for UHR is 238.54.

**Table 3 Univariate analysis of UHR and carotid intima-media thickening in the subgroups.**

| Item | OR (95% CI) | P |
|---|---|---|
| Gender: | | |
| Male | 1.002 [1.000–1.004] | 0.013 |
| Female | 1.000 [0.996–1.003] | 0.783 |
| Age groups: | | |
| <60 | 1.003 [0.997–1.008] | 0.349 |
| ≥60 | 1.002 [1.001–1.004] | 0.001 |
| Gender and age groups: | | |
| Male <60 years | 1.000 [0.993–1.006] | 0.889 |
| Male ≥60 years | 1.003 [1.001–1.004] | 0.004 |
| Female <50 years | — | |
| Female ≥50 years | 1.001 [0.999–1.002] | 0.535 |

**Table 4 Multifactorial analysis of UHR and carotid intima-media thickening in elderly men.**

| Item | OR (95% CI) | P |
|---|---|---|
| UHR | 1.002 [1.000–1.004] | 0.028 |
| Diabetes | 1.85 [1.05–3.27] | 0.033 |
| BMI | 1.05 [0.98–1.13] | 0.179 |
| Differential pulse pressure | 1.03 [1.01–1.04] | <0.001 |
| TC | 1.04 [0.71–1.52] | 0.829 |
| LDL-C | 1.47 [0.94–2.29] | 0.091 |

**Note:** TC, LDL-C are in mmol/L.

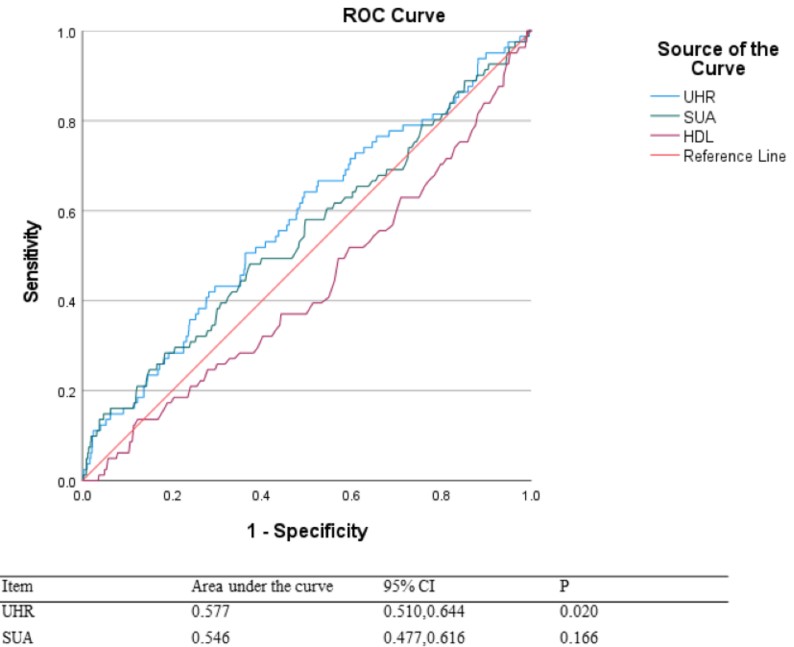

| Item | Area under the curve | 95% CI | P |
|---|---|---|---|
| UHR | 0.577 | 0.510,0.644 | 0.020 |
| SUA | 0.546 | 0.477,0.616 | 0.166 |
| HDL | 0.432 | 0.365,0.499 | 0.041 |

**Figure 2 ROC analysis.**

## DISCUSSION

The primary objective of this study was to investigate the association between the serum uric acid to high-density lipoprotein cholesterol ratio (UHR) and carotid intima-media thickness (cIMT) in a rural Chinese population. We aimed to determine whether UHR could serve as a reliable marker for early detection and intervention of carotid atherosclerosis, and to explore how this association might vary across different age and sex groups. In this study, we established that UHR is a risk factor for increased cIMT. Specifically, higher UHR values were significantly associated with thicker cIMT. UHR, age, hypertension, diabetes, smoking, alcohol consumption, SBP, pulse pressure (PP) difference, GLU, TC, LDL-C, and SUA were all identified as significant risk factors for cIMT. Gender was the only factor negatively correlated with cIMT. We found that the association between UHR and carotid intima-media thickening was particularly pronounced in older men. Moreover, our study suggests that the combined metric of UHR provides additional predictive value. The use of UHR as a combined marker may offer a more nuanced understanding of the complex interplay between metabolic and inflammatory pathways in the development of atherosclerosis.

The relationship between the UHR and carotid atherosclerosis has not been previously explored. Several studies have highlighted the roles of SUA and HDL-C in cardiovascular diseases. A recent meta-analysis supported a significant correlation between elevated SUA levels and increased cIMT, suggesting SUA's role in promoting atherosclerosis through oxidative stress and inflammation (*Ma et al., 2021*). Another cross-sectional study demonstrated a weak positive correlation between SUA and cIMT in obese children, indicating that SUA may influence lipid metabolism and thereby affect the carotid intima (*Serret-Montoya et al., 2023*). Additionally, research on a middle-aged population found that normal-range SUA levels were associated with inflammation and arterial stiffness, implicating hyperuricemia in a pro-inflammatory state (*Laučytė-Cibulskienė et al., 2022*). Studies also showed a negative association between HDL-C and coronary heart disease (CHD) and cIMT, highlighting HDL-C's protective effects through anti-inflammatory and antioxidative mechanisms (*Gordon et al., 1989*; *Takase et al., 2023*).

Our study contributes to this body of knowledge by demonstrating that UHR, as a combined marker, provides additional predictive value for cIMT. Unlike previous studies focusing on SUA and HDL-C individually, our findings reveal that higher UHR values are significantly associated with increased cIMT. This association is particularly pronounced in older men, suggesting that UHR offers a more nuanced understanding of the complex interplay between metabolic and inflammatory pathways in atherosclerosis development.

The potential mechanisms underlying these findings could involve the combined effects of SUA and HDL-C on oxidative stress and inflammation. Elevated SUA levels can lead to endothelial dysfunction and promote inflammatory processes, which are critical in the development of atherosclerosis (*Kimura, Tsukui & Kono, 2021*). On the other hand, HDL-C provides protective effects through its antioxidative and anti-inflammatory properties. Therefore, a high UHR, reflecting both elevated SUA and decreased HDL-C levels, could indicate a state of heightened oxidative stress and reduced protective capacity,

leading to greater carotid intima-media thickenin (*Linton et al., 2023*; *Rye & Barter, 2014*; *Barter et al., 2004*).

This study conducted ROC analysis to evaluate the relationship between UHR and the risk of carotid intima-media thickening, with results indicating that UHR has certain predictive value. In clinical practice, UHR values can be used for prevention and intervention to reduce the risk of cardiovascular and cerebrovascular diseases. However, is it universally applicable in all scenarios? The study reveals a U-shaped relationship between plasma HDL-C levels and the risk of cardiovascular and cerebrovascular diseases, indicating that excessively high HDL-C levels may still increase the risk of these conditions (*Franczyk et al., 2021*). European guidelines suggest elevated HDL-C levels (>90 mg/dL) have been associated with an increased risk of atherosclerotic cardiovascular disease (ASCVD) (*Mach et al., 2019*). Plasma HDL-C may become dysfunctional in individuals with cardiovascular and cerebrovascular diseases under conditions of oxidative stress or inflammation. In such cases, UHR loses its predictive value for carotid intima-media thickening.

There were several limitations in this study. Firstly, the cross-sectional design of the study limits the ability to establish causality between UHR and cIMT. As a result, while we observed significant associations, we cannot infer a direct cause-and-effect relationship. Longitudinal studies are necessary to confirm the causal link between UHR and cIMT and to observe the progression of atherosclerosis over time. Secondly, the study population was limited to rural residents of Jizhou District, Tianjin, China. This geographic and demographic restriction may affect the generalizability of our findings to other populations. Rural populations may have different lifestyle factors, dietary habits, and healthcare access compared to urban populations, which could influence the observed associations. Future studies should include diverse populations from different regions to enhance the generalizability of the results. Thirdly, the reliance on self-reported data for lifestyle factors such as smoking and alcohol consumption introduces the potential for recall bias. Participants may underreport or overreport their behaviors, leading to misclassification and affecting the accuracy of the associations observed. Improved data collection methods, such as biomarker validation of self-reported behaviors, could mitigate this issue in future research. Additionally, while we adjusted for several potential confounders, residual confounding may still be present. Unmeasured variables, such as dietary intake, physical activity levels, and genetic factors, could influence the relationship between UHR and cIMT. Most importantly, diet is a critical factor influencing uric acid levels. However, this study did not qualitatively or quantitatively examine the impact of diet on uric acid in the target population. Existing research indicates that a high-fructose diet can induce hyperuricemia (*Lubawy & Formanowicz, 2023*). Therefore, future studies should quantitatively assess the effects of dietary factors on uric acid. Comprehensive data collection could be employed in future studies to better account for these confounders.

## CONCLUSION

This study demonstrates that the UHR is a significant predictor of cIMT and carotid intima-media thickening, particularly in older men. Our findings suggest that UHR could

serve as a valuable marker for early detection and intervention of carotid atherosclerosis. The identification of UHR as a risk factor for carotid atherosclerosis provides a new tool for assessing their cardiovascular risk. By monitoring UHR levels, patients, especially older men, can gain insights into their risk of developing atherosclerosis and take proactive steps to manage their health. This could involve lifestyle modifications, such as improving diet and increasing physical activity, as well as medical interventions aimed at lowering uric acid levels and managing cholesterol. Early intervention targeting uric acid or HDL-C levels could serve as a beneficial strategy to mitigate carotid intima-media thickening risk. By utilizing UHR, physicians can identify high-risk patients earlier and implement targeted prevention strategies. This could lead to better patient outcomes, reduced incidence of cardiovascular events, and more efficient use of healthcare resources. Early detection and management of carotid atherosclerosis can reduce the burden of stroke and other cardiovascular diseases, leading to lower healthcare costs and increased productivity.

## ACKNOWLEDGEMENTS

We thank all participants of the Tianjin Brain Study, and local medical care professionals for their valuable contributions.

### Funding

This study was sponsored by Tianjin Health Research Project (No. TJWJ2022ZD012). The funders had no role in study design, data collection and analysis, decision to publish, or preparation of the manuscript.

### Grant Disclosures

The following grant information was disclosed by the authors:
Tianjin Health Research Project: TJWJ2022ZD012.

### Competing Interests

The authors declare that they have no competing interests.

### Author Contributions

- Guibao Luo performed the experiments, analyzed the data, prepared figures and/or tables, and approved the final draft.
- Haiying Wang performed the experiments, analyzed the data, prepared figures and/or tables, and approved the final draft.
- Yajun Cao performed the experiments, analyzed the data, prepared figures and/or tables, and approved the final draft.
- Juan Hao performed the experiments, prepared figures and/or tables, and approved the final draft.
- Taofeng Tan performed the experiments, prepared figures and/or tables, and approved the final draft.

- Yingzhe Shao performed the experiments, prepared figures and/or tables, and approved the final draft.
- Xianjia Ning conceived and designed the experiments, authored or reviewed drafts of the article, and approved the final draft.
- Chunsheng Yang performed the experiments, authored or reviewed drafts of the article, and approved the final draft.
- Jinghua Wang conceived and designed the experiments, authored or reviewed drafts of the article, and approved the final draft.
- Yan Li conceived and designed the experiments, authored or reviewed drafts of the article, and approved the final draft.

## Human Ethics

The following information was supplied relating to ethical approvals (*i.e.*, approving body and any reference numbers):

Ethics Committee of the General Hospital of Tianjin Medical University (approval number: IRB2018-100-01).

## Data Availability

The raw measurements are available in the Supplemental Files.

## Supplemental Information

Supplemental information for this article can be found online at http://dx.doi.org/10.7717/peerj.20053#supplemental-information.

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
