# Peer review of "Uric acid to HDL cholesterol ratio as a novel predictor of carotid intima-media thickness: a cross-sectional study in rural China"

_PeerJ, doi:10.7717/peerj.20053_

## Round 0.1 · original submission · Major Revisions

**Language Note:** When you prepare your next revision, please either (i) have a colleague who is proficient in English and familiar with the subject matter review your manuscript, or (ii) contact a professional editing service to review your manuscript. PeerJ can provide language editing services - you can contact us at [email protected] for pricing (be sure to provide your manuscript number and title). – PeerJ Staff

Reviewer 1 ·

Basic reporting

An interesting article has been submitted for my evaluation.

Introduction - The subject is a good introduction, but I doubt the specific issues.

To start with, why do researchers refer only to selected issues, as stroke, and not, for example, to heart attacks (?) or more general descriptions of the consequences of advanced atherosclerosis (?).

I also have reservations about the unambiguous formulation of HDL-C. Today, we know about the dual nature of the HDL molecule, which in the group of patients with cardiovascular disease, is subject to the impact of oxidative stress and inflammation, and the HDL becomes dysfunctional - researchers should at least mention this.

The aim of the article is formulated correctly.

The authors incorrectly describe chronic renal failure (CKD > >= 4) in the exclusion criteria; they need to know that only CKD stage 5 = chronic renal failure. Hence, writing chronic kidney disease instead of chronic renal failure would be better.

The authors misuse the notation LDL and HDL - they did not evaluate whole lipoproteins, but cholesterol in these lipoproteins; therefore, "c" should be added, i.e., it should not be low-density lipoprotein but low-density lipoprotein cholesterol (LDL-C), not high-density lipoprotein but high-density lipoprotein cholesterol (HDL-C). For example, in lines 176, 184, 215, Tables and Figures, etc.

In Table 1, it is not known what the notation in brackets means. It is most likely the percentage share in a given group - this should be explained in the description.

I wonder about the division by age into those <60. This is particularly important in the context of women, because here, from the age of 50, menopause has an impact. Hence, it may be better to analyse those <50.
In the table description, the abbreviation "UHR" should be explained.

In lines 146 and 146, there is an error in references, which should be corrected.
In lines 115-151, the units for the BMI parameter should be added.

I suggest that in Table 1, two groups (males and females) be compared and the statistical significance of "p" assessed.

Throughout the article, the literature is incorrectly cited; instead of numbers, there is "0" everywhere - this should be corrected.

Some sentences are unclear, e.g., "Our findings suggest that UHR could serve as a valuable marker for early detection and intervention of carotid atherosclerosis." I do not understand what the authors mean here by marker for intervention (?)

Conclusions are fine.
In the Limitations, they should remember that they haven't assessed the impact of diet, which has a crucial place in uric acid values. I want to point out here that not only does a high-purine diet have an effect, but e.g., a diet high in fructose, see https://doi.org/10.3390/ijerph20043596.

Experimental design

The design of the experiment is fine. I do not have any ethical concerns. Everything is clear here.

Validity of the findings

The article is quite novel, but the aspects of the dual nature of HDL need to be better elaborated.

Reviewer 2 ·

Basic reporting

The association between the uric acid-to-high-density lipoprotein cholesterol ratio (UHR) and carotid intima-media thickness (IMT) is a novel research topic, with no prior studies directly addressing this relationship in the existing literature.

1. The citation reference numbers in this paper appear to be incorrectly linked or non-functional. Please review and correct the reference formatting.

2. Regarding the study population, did the participants have a prior history of stroke? If so, what was the prevalence of stroke history in the cohort?

3. The authors used univariate and multivariate linear regression models to evaluate the association between UHR and IMT. However, was the distribution of IMT assessed for normality? Additionally, were the assumptions of the linear regression model (e.g., linearity, homoscedasticity, normality of residuals) evaluated and met?

4. The overall ROC analysis suggests a modest predictive value of UHR for carotid intima-media thickening. This should be discussed further in terms of clinical relevance and limitations.

5. Please review prior literature on UHR. What other biomarkers or diseases have been previously associated with UHR?

6. All analyses used UHR as a continuous variable. Do the authors recommend a specific cut-off value for UHR to facilitate clinical categorization (e.g., high vs. low UHR) for potential interventions?

Experimental design

-

Validity of the findings

lines 248-250: Longitudinal studies are necessary to confirm the causal link between UHR and cIMT and to observe the progression of atherosclerosis over time.

---

## Round 0.2 · accepted · Accept

Thank you for addressing all of the reviewer comments. Your manuscript is now ready for publication.

Reviewer 1 ·

Basic reporting

no comment

Experimental design

no comment

Validity of the findings

no comment

Additional comments

no comment